# OpenReview forum: "Causal Abstraction Finds Universal Representation of Race in Large Language Models"
_ICLR.cc/2025/Conference — Submitted to ICLR 2025_

### Official Review · Reviewer_CDow · 2024-10-22

**Soundness:** 3
**Presentation:** 3
**Contribution:** 1
**Rating:** 5
**Confidence:** 3

**Summary:**

The authors collected dataset on college admissions and hiring, and applied causal tracing to learn rotation and masking matrices to perform interventions (regarding demographic information) mechanistically on the model. The learned masks for race seem to be applicable to perform the intervention across tasks.

**Strengths:**

The paper is well-presented, with a clear explanation of the causal abstraction method. The construction of the hiring and college admission datasets for examining racial bias is a valuable contribution to the field.

**Weaknesses:**

My primary concern is that the work appears to be a straightforward application of existing causal abstraction techniques to a new dataset, followed by an interpretation of the results. While the analysis of the findings is thorough, the overall contribution seems limited.

**Questions:**

See weaknesses above.

---

### Official Review · Reviewer_mav4 · 2024-10-29

**Soundness:** 2
**Presentation:** 3
**Contribution:** 2
**Rating:** 3
**Confidence:** 3

**Summary:**

This paper adopts the idea of causal abstraction to identify the racial bias in LLM models, in the context of two high-stakes decision settings: college admissions and hiring.  A large amount of experiments have been conducted to show the performance of the proposed approach and the work provides the evidence for a universal mechanism of racial bias in LLM's decision-making.

**Strengths:**

1. The description of method is clear.
2. A comprehensive simulation study has been conducted.

**Weaknesses:**

1. From my perspective, this work is primarily "applied," using existing methods to analyze certain "properties" of LLMs without demonstrating substantial novelty.
2. This works focus on college admissions and hiring, which may be too narrow in scope, raising questions about the generalizability of its findings.

**Questions:**

1. The paper claims that it is the first study to apply causal abstraction in high-stakes decisions. However, the authors do not provide enough illustrations on the difficulties of such application. For instance, what is the unique challenge compared to the prior study and how to handle these unique challenges?
2. It will be better to provides more details about the ``causal model" in the main context in the end of Section 2.2, as it seems to be an reference in the learning task.
3. The paper claims that it identifies causal representations of race information. It will be better to see more interpretations about the learned representations and why it is being considered as ``causal" representations.

---

### Official Review · Reviewer_DAPS · 2024-11-02

**Soundness:** 3
**Presentation:** 3
**Contribution:** 3
**Rating:** 6
**Confidence:** 4

**Summary:**

This paper utilizes causal abstraction to investigate racial bias in the decision-making of LLMs. By aligning causal models with neural activations through distributed alignment search (DAS), the proposed method offers a mechanistic understanding of how LLMs encode these biases and how they affect decisions in contexts such as university admissions and hiring. The method yields high transfer interchange intervention accuracy (IIA), demonstrating its reliability in identifying causal representations, outperforming other techniques such as probing. Additionally, it provides evidence for a universal mechanism of racial bias in LLMs' decision-making.

**Strengths:**

1. The targeted problem is valid and important, as LLMs are increasingly employed in hiring and admissions decisions, making it crucial to address the biases encoded within these models.
2. The application of causal abstraction to LLMs' high-stakes decision-making is novel, and the effectiveness of the proposed method is demonstrated through several well-designed tasks.
3. The paper is well-written. The result in Section 3.2, where DAS identifies that the race representation found in the ADMISSIONS dataset aligns with the model’s general decision-making process, is particularly interesting. Additionally, Section 3.3 offers valuable insights into how race is encoded in LLMs and how interventions influence the outcomes.

**Weaknesses:**

See my detailed questions below.

**Questions:**

1. I am confused about the sample sizes in the training and test sets differing among the Alpaca, Mistral, and Gemma models.
2. I recommend using a more symbolic and standardized mathematical language to describe interventions on neural networks, which would enhance readability.
3. If I have understood correctly, the ADMISSIONS and HIRING datasets are synthetic, generated from some human-defined causal structures. If these causal structures are similarly designed, it seems reasonable that the tasks share a universal representation of race. However, whether this conclusion holds for real-world data remains questionable, particularly given the unknown nature of true causal structures.

---

### Official Review · Reviewer_bZfu · 2024-11-03

**Soundness:** 1
**Presentation:** 2
**Contribution:** 2
**Rating:** 3
**Confidence:** 2

**Summary:**

This paper uses causal abstraction to investigate how models process racial information in two critical decision-making areas: college admissions and hiring. The goal is to address potential biases in large language models. The authors use Distributed Alignment Search (DAS) to find alignments between a causal model and neural activations and apply Distributed Interchange Intervention (DII) to replace the race variable for counterfactual comparisons. They evaluate the outcomes with Interchange Intervention Accuracy (IIA). The results show that DAS can identify neurons that consistently affect prediction outcomes in admissions, performing better than probing methods and yielding strong results in hiring tasks as well.

**Strengths:**

1 The paper uses causal abstraction to find causal representations based on counterfactual outcomes related to race in admissions decisions. It also introduces a new approach for performing interchange interventions in other tasks, using a custom dataset focused on college admissions and hiring.
2  The authors test their DAS model by comparing IIA results with those of a probing model and a random model, showing that their method has better accuracy and reliability.
3  The proposed method is easy to apply to tasks like admissions and hiring, suggesting that race representations could be consistent across different tasks.

**Weaknesses:**

1. The counterfactual examples generated by the authors appear to change only the sensitive attributes, without adjusting related information. This approach may lead to unrealistic counterfactual examples. Additionally, it would be beneficial if the authors discussed how to identify sensitive attribute information within the representation space.

2.  The method presented requires further clarification in some parts. For example, in Figure 2, both race information and GPA change simultaneously. This makes it difficult to directly attribute the change in the outcome to model discrimination, which does not align with the concept of counterfactual fairness.

3.  Some notations in the paper require further clarification, as certain symbols in the formulas are ambiguous. For instance, the symbol connecting "Mask" and "RF" in Formula 1 is unclear, as is the reference to M_{\text{mask}}. In key formulas like Formula 1 and Formula 3, the concept of "mask" may be difficult for readers to grasp. Providing a precise definition of "mask" would enhance understanding.

4.  The paper should more clearly explain the relationship between "base input" and "source input," which are crucial elements of the dataset. In the "Interchange Intervention" section, for example, the connection between $b$ (base input) and $s$ (source input) lacks detail, making it difficult to prevent mismatches that could impact causal inference. A clearer explanation of the one-to-many relationship between these inputs would improve reliability. For instance, in line 188, where $S = {s_i}^k$ represents source inputs and $b$ is a base input, it is unclear whether $b$ corresponds one-to-one or shares a common base.

**Questions:**

Please check the weakness.

---

### Official Review · Reviewer_9Ymb · 2024-11-03

**Soundness:** 3
**Presentation:** 2
**Contribution:** 2
**Rating:** 5
**Confidence:** 3

**Summary:**

This paper applies causal abstraction to explore how LLMs encode and use race information in high-stakes decision-making tasks, specifically in college admissions and hiring scenarios. Using models like Alpaca 7B, Mistral 7B, and Gemma 2B, the authors identify racial representations within neural activations and examine their impact on model decisions. In addition to a few standard tasks evaluated via IIA, the authors introduce a new task, RaceQA, to validate these representations and investigate whether racial encoding can directly influence model outputs. The authors also created new datasets, including college admissions and hiring tasks, which is a nice addition to the literature.

**Strengths:**

The paper takes an established method, causal abstraction, and applies it to investigate racial bias within LLMs in high-stakes domains like college admissions and hiring. While this application is relatively new, it remains primarily empirical rather than methodologically innovative, as it does not introduce novel techniques or substantial modifications to existing methods. The originality here lies mainly in the domain application rather than in advancing causal abstraction methodology itself (as I will comment below, this may limit it impact in more foundational/methodological contributions to machine learning).

From the empirical examination perspective, the paper is quite thorough in its experimental setup, testing racial representations across several models and settings. The inclusion of the RaceQA task adds rigor to the analysis by enabling targeted examination of racial bias.  The paper is generally well written, while the clarity could be improved by providing more background of applying causal abstraction to LLMs.

In summary, the significance of this work mainly lies in its empirical findings regarding racial bias in LLMs, which is relevant for fairness and interpretability research in AI. The paper’s contribution to understanding how race is encoded in models used for decision-making tasks aligns with the growing importance of responsible AI.

**Weaknesses:**

1. While the paper applies causal abstraction to high-stakes decision tasks in LLMs, it does not introduce new methodological advancements or improvements to the causal abstraction framework itself. The core method is derived from prior work such as Geiger et al. (2023). This limits the paper’s novelty and impact, especially for a conference venue that focuses on advancing foundational ML methods.

Given its focus within the applied contexts, this work may find a better fit in conferences or journals that emphasize fairness/ethics in applications, rather than a venue like ICLR that prioritizes foundational ML innovations.

2. The empirical findings on racial bias in LLMs are valuable, but the paper lacks a detailed discussion on how these insights could be applied to mitigate bias in real-world applications. I understand that the investigation presented in this paper is a first step, but to enhance the practical relevance of the work, the authors could include a section discussing actionable steps for reducing racial bias in models based on the causal insights gathered, or suggest techniques for using causal interventions in training or fine-tuning to limit the influence of biased representations.

3. The paper focuses on two high-stakes decision tasks: college admissions and hiring. While these are relevant domains, they may not fully capture the breadth of racial bias issues across varied applications of LLMs. Additionally, causal representations found in these tasks may not generalize to domains with different types of sensitive information.

4. As the authors mentioned "However, due to the complexity of large language models, success in changing the output on a decision task does not necessarily imply that the change happened because we have changed the race." Causal interventions in high-dimensional neural spaces can be prone to noise and instability. It would nice if the the authors could test how stable and consistent the causal representations are under various conditions (e.g., across different LLM architectures, hyperparameters, or data distributions).

5. Since model bias is often influenced by training data, it would be beneficial to discuss any inherent biases in the datasets that could affect the findings, as well as any limitations in using synthetic versus real-world data. Additionally, a more detailed description of how the RaceQA dataset was constructed, particularly around name selection and race labels, would provide transparency and help evaluate the dataset’s validity.

**Questions:**

1. Could the authors elaborate on whether any adaptations or refinements were made to the causal abstraction method specifically for application to LLMs? If not, do the authors see any specific areas where causal abstraction could be further developed to better capture the complexities of high-dimensional language models? Discuss along this direction could help strengthen the methodological contributions of the paper.

2. How do the authors envision that their findings could be used to mitigate racial bias in LLMs, especially in practical, real-world settings? Are there specific steps that developers could take based on this research?

3. Did the authors consider testing causal representations across additional domains, such as criminal justice or finance, where decision-making may also be influenced by sensitive information? Could they provide insights into whether the racial representations found in admissions and hiring tasks are expected to generalize to other high-stakes contexts?

4. Could the authors elaborate on the selection criteria for names and races used in RaceQA, as well as any potential biases introduced by these choices? How do they ensure the synthetic data accurately reflects real-world demographic distributions?


7. Given that causal abstraction identifies race representations in admissions and hiring tasks, how do the authors view the generalization of these findings to other sensitive attributes, such as gender or age? Are there any foreseeable challenges in extending their framework to such attributes?

**Details Of Ethics Concerns:**

The creation of the dataset lacks sufficient details, except limited details on the prompts used in Appendix A. It could encode/introduce inherent bias during the data creation.

---

### Official Review · Reviewer_fvqN · 2024-11-04

**Soundness:** 1
**Presentation:** 2
**Contribution:** 1
**Rating:** 3
**Confidence:** 5

**Summary:**

The paper explores the potential application of the previous work on causal abstraction (Geiger et al., 2023) and attempts to see if there is a representation of race in large language models. The work builds heavily upon previous works (Geiger et al., 2023, Wu et al., 2024) and considers three models on two synthetic settings, where several variables are transformed from the tabular format into natural language descriptions.

**Strengths:**

The strength of the paper comes from the attempt to explore the universal representation of protective feature in LLM reasoning. The question of interest is important and timely.

**Weaknesses:**

(1) The paper relies heavily on results from previous works, and the paper is more of a validation of existing claims

For instance, readers could refer to previous works to get full details of the presented theory (lines 160 -- 161).

(2) The scope of the empirical study is rather limited, and the claim is not well-justified

The experiments are conducted on two synthetic data sets, which are conversions of tabular data to natural language, with no more than five variables as placeholders (Table 1).  It is hard to justify the claim that universal representation exists, based on the presented empirical evaluations.

**Questions:**

How applicable is the current claim:

- if the model size increases?

- if the language data is not synthetic and not converted from tabular data?

---

### Meta-Review · Area_Chair_fgZw · 2024-12-21

**Metareview:**

This paper applied causal abstraction (Geiger et al., 2023) to explore how LLMs encode and use race information in high-stakes decision-making tasks, specifically in college admissions and hiring scenarios. Using models like Alpaca 7B, Mistral 7B, and Gemma 2B, the authors identify racial representations within neural activations and examine their impact on model decisions. While the paper applies causal abstraction to high-stakes decision tasks in LLMs, the work builds heavily upon previous works (Geiger et al., 2023, Wu et al., 2024) and does not introduce new methodological advancements or improvements to the causal abstraction framework itself. The reviewers have common concerns on the limited novelty, contribution, as well as the empirical study, which unfortunately were not addressed at all during rebuttal. After discussion with the reviewers, we agreed it is not quite ready for publication.

**Additional Comments On Reviewer Discussion:**

All reviewers have raised similar concerns on this paper ref the limited novelty, contribution, as well as the empirical study. While the paper applies causal abstraction to high-stakes decision tasks in LLMs, the work builds heavily upon previous works (Geiger et al., 2023, Wu et al., 2024) and does not introduce new methodological advancements or improvements to the causal abstraction framework itself. The authors have not responded to this criticism and we agreed it is not quite ready for publication.

---

### Decision · Program_Chairs · 2025-01-22

Reject